# Network heterogeneity regulates steering in actin-based motility

Rajaa Boujemaa-Paterski[1], Cristian Suarez[1], Tobias Klar[1], Jie Zhu[2], Christophe Guérin[1], Alex Mogilner[2], Manuel Théry[1,3] & Laurent Blanchoin[1,3]

The growth of branched actin networks powers cell-edge protrusions and motility. A heterogeneous density of actin, which yields to a tunable cellular response, characterizes these dynamic structures. We study how actin organization controls both the rate and the steering during lamellipodium growth. We use a high-resolution surface structuration assay combined with mathematical modeling to describe the growth of a reconstituted lamellipodium. We demonstrate that local monomer depletion at the site of assembly negatively impacts the network growth rate. At the same time, network architecture tunes the protrusion efficiency, and regulates the rate of growth. One consequence of this interdependence between monomer depletion and network architecture effects is the ability of heterogeneous network to impose steering during motility. Therefore, we have established that the general principle, by which the cell can modulate the rate and the direction of a protrusion, is by varying both density and architecture of its actin network.

[1] CytomorphoLab, Biosciences & Biotechnology Institute of Grenoble, Laboratoire de Physiologie Cellulaire & Végétale, Université Grenoble-Alpes/CEA/CNRS/INRA, 38054 Grenoble, France. [2] Courant Institute of Mathematical Sciences and Department of Biology, New York University, New York, NY 10012, USA. [3] CytomorphoLab, Hôpital Saint Louis, Institut Universitaire d'Hematologie, UMRS1160, INSERM/AP-HP/Université Paris Diderot, 75010 Paris, France. Rajaa Boujemaa-Paterski, Cristian Suarez and Tobias Klar contributed equally to this work. Correspondence and requests for materials should be addressed to A.M. (email: mogilner@cims.nyu.edu) or to M.T. (email: manuel.thery@cea.fr) or to L.B. (email: laurent.blanchoin@cea.fr)

Cell migration is an evolutionary conserved mechanism, essential for the proper development of living organisms[1]. A fundamental and still open question in biology is how cells direct their migration in response to external signals[2, 3]. Much effort has been focused on understanding the mechanism of the first step in this process: membrane protrusion and its regulation[4]. Actin polymerization produces the intracellular force[5] that protrudes a thin and flat structure, called lamellipodium, which borders the leading edge of a motile cell over tens of micrometers[6-8]. The lamellipodial actin is a densely branched and dynamic meshwork[9-11]. Near the cell membrane, sustained Arp2/3-mediated dendritic nucleation[12], filament assembly and

**a** Restricted bar — 0 min, 10 min, 20 min

**b** Restricted dot — 0 min, 10 min, 20 min

**c** Growth rate (μm s⁻¹ μM⁻¹ available [actin])
A: 0.005 ± 0.001
B: 0.019 ± 0.004
****

**d** Unrestricted bar — 0 min, 13 min, 26 min

**e** Unrestricted dot — 0 min, 8 min, 17 min

**f** Single filament — 0 min, 5 min, 10 min

**g** Growth rate (μm s⁻¹ μM⁻¹ available [actin])
D: 0.029 ± 0.003
E: 0.033 ± 0.003
F: 0.030 ± 0.003
n.s.

**h** Reconstituted lamellipodium — Polymerization, Growth, $F_{push}$, $F_{pull}$

$$\underbrace{V}_{\text{Growth rate}} \approx \underbrace{k_{on}}_{\substack{\text{Monomer}\\\text{assembly}\\\text{rate}}} \times \underbrace{\delta}_{\substack{\text{Monomer}\\\text{size}}} \times \underbrace{c \times C_0}_{\substack{\text{Monomer}\\\text{depletion factor}}} \times \underbrace{\Phi}_{\substack{\text{Geometrical/}\\\text{mechanical}\\\text{factor}}}$$

disassembly of the lamellipodial actin network are finely tuned in space and time through coordinated activities of regulatory factors[13, 14]. Collectively, these processes generate cohesive branched actin networks[9, 11, 15], along the leading edge that expand locally leading to directed motility in response to environmental cues[6, 7, 16, 17]. Steering during motility is tightly linked to regulation of the Arp2/3-branching activity[18]. However, how actin-network organization and growth regulates steering is unclear[3].

In vitro reconstituted propulsion of bacteria, viruses or small particles brought insights on how a minimal set of two molecular activities—Arp2/3 complex-driven nucleation and barbed-end capping by capping proteins—can result in the growth of protrusive actin organization[19–25]. The surface density of the nucleation-promoting factors (NPFs), the size and shape of the motile particles and the viscosity of the medium affect the velocity of propulsion[21, 26, 27]. In addition, a growing actin network is a mechanosensitive system that can respond and adapt to external forces[28]. However, we know little about how actin polymerization defines the rate of growth of a branched actin ultrastructure pushing against a load.

Here, we asked how the architecture of a branched actin network affects its growth and investigated the key parameters controlling speed and steering during motility. To achieve a high precision in controlling the organization of a growing branched actin network, we developed a methodology that combined contactless micropatterning of variable concentration of NPFs[29, 30], with an in vitro reconstituted actin-based motility assay[24]. Using this approach, we generated a diversity (in terms of size (geometry) and NPF concentration) of nucleation areas and studied their impact on the growth of branched actin networks. At the same time, we used quantitative fluorescence imaging to determine the density of the branched actin network and its relationship with network growth behavior. To explain the growth rate of the actin network, we developed a mathematical model relying on minimal assumptions. The model revealed that the local actin-monomer concentration at the site of active nucleation and the architecture of the branched network are the two fundamental parameters controlling motility in our experimental system. Our model was validated by a series of experiments where the growth behavior of the actin network was modulated by the geometry, density and composition of the nucleation area. In agreement with the model predictions, we reconstituted controlled steering of heterogeneous actin networks using NPFs patterned at a submicrometer scale. Therefore, the fine-tuning of only two parameters was sufficient to fully recapitulate the observed growing behavior of a branched actin network.

## Results

**Parameters controlling the actin network growth rate.** To investigate how the organization of actin filaments modulates actin-based motility, we reconstituted in vitro branched actin networks with a diversity of nucleation geometries and characterized their growth dynamics. We assembled actin networks on functionalized micropatterned surfaces uniformly coated with NPFs, in the presence of a defined set of purified cellular factors (Fig. 1a–e, Supplementary Figs 1 and 2 and refs 28, 30, 31). We imaged fluorescently labeled actin to follow actin-network assembly. This novel versatile method allowed actin assembly at a nucleation site and the growth of actin filaments at their barbed ends to be geometrically constrained (see two-color experiments Supplementary Fig. 1a), which in turn induced the growth of a cohesive actin network restricted in the extent of his growth by the presence of capping proteins (Supplementary Figs 1b and 2a, b). Hence the method was used to assemble thin and flat Arp2/3-generated lamellipodium-like structures thereafter referred to as "LMs" (Supplementary Fig. 2a, b compared LMs with a classical bead comets Supplementary Fig. 2c, d).

Because the geometry of the nucleation sites could be altered with the patterning method, we compared the configuration of LMs from a functionalized NPF-bar-shaped pattern of $3 \times 15 \, \mu m^2$ (Fig. 1a, c and Supplementary Movie 1) and thin-tail branched networks from NPF-spots of $<1 \, \mu m^2$ (Fig. 1b, c and Supplementary Movie 1). Interestingly, the growth rate of the "restricted" networks (i.e. a restriction imposed by the presence of capping proteins) varied with the geometry of the nucleation area. LMs from 15-μm bars grew significantly slower than those from small spots (Fig. 1c). This difference was not due to a dependency of actin assembly on the geometry of the nucleation area because, regardless of the nucleation area and its geometry (bar vs. dot, Fig. 1d, e and Supplementary Movie 2), with "unrestricted" actin networks (i.e. in the absence of capping protein), the network growth rate was not statistically different from the rate measured for individual (free) actin filaments (Fig. 1f, g).

The rate of free actin filament elongation was expressed by the canonical equation for actin filament elongation, whereby, $V_0 = k_{on} \times \delta \times C$, where $k_{on} \approx 10 \, \mu M^{-1} \, s^{-1}$ is the polymerization rate constant, $\delta \approx 0.003 \, \mu m$ is the half-size of actin monomer, and $C$ is the local actin monomer concentration (Supplementary Methods[32]). This equation predicted a free polymerization rate $\approx 0.03 \, \mu m \, \mu M^{-1} \, s^{-1}$, which was in agreement with the measurements for single filaments and unrestricted networks. In comparison to single filament assembly, quantitative analysis of the restricted network revealed a 6- and 1.6-fold decrease in

---

**Fig. 1** Nucleation geometry controls the actin network growth rate. **a, b** The growth of different restricted actin organizations. **a** Actin network emerging from $3 \times 15 \, \mu m^2$ GST-pWA-coated bar. Conditions: 6 μM G-actin Alexa-568 labeled, 18 μM Profilin, 120 nM Arp2/3 complex, 25 nM CP. **b** Same as **a** but for GST-pWA-coated sub-micron dot. **c** The growth rate of actin networks was calculated in μm s⁻¹ μM⁻¹ of available [G-actin]. *Red bars* represent mean speed values. *Error bars* show mean s.d. for $n = 20$ LMs from four experiments **a** and $n = 13$ LMs from five experiments **b**. ****$p < 0.0001$, multiple comparison Šídák method. **d, e** The growth of different unrestricted actin networks. **d** Actin assembly from $3 \times 15 \, \mu m^2$ GST-pWA-coated bar. Conditions: 1 μM G-actin Alexa-568 labeled, 3 μM Profilin, 80 nM Arp2/3 complex **e** same as **d** but for $1 \times 1 \, \mu m^2$ GST-pWA-coated dot. **f** The assembly of single actin filaments. Conditions: 1 μM G-actin Alexa-568 labeled, 3 μM Profilin. In **f** each filament is identified by a *star*. In **a**, **b**, **c** and **d**: *purple* shows the nucleation sites; in **b**, speckles in the comet tail indicated by *colored stars* were used for speed measurement. **g** For homogeneity and for each condition, the growth rate of actin filaments or actin organizations was calculated in μm s⁻¹ μM⁻¹ of available G-actin. *Red bars* represent mean speed values provided above. *Error bars* show mean s.d. for *n* patterns per condition, $n = 26$ from four experiments **d**, $n = 11$ from three experiments **e** and $n = 26$ from four experiments **f**. Adjusted *p* values were computed according to Šídák method to assess statistically significant difference in mean growth rates measured for each condition. **h** Cartoon for the reconstituted flat LMs: LMs grow from NPF-coated patterns (*light red rectangle*) printed on Silane-PEG-coated slide (*darkened glass*). The height between glass and coverslip was controlled by two pieces of calibrated tape (*yellow walls*). Schematic of actin assembly and the parameters used in the model equation: Network growth rate "*V*" depends on (i) the barbed end rate of assembly ($k_{on}$), (ii) the monomer size ($\delta$), (iii) the local monomer concentration ($C$, factor 1) at the nucleation site, and (iv) a geometry/mechanical factor $\Phi$ that integrates the pulling force (*black arrows*, factor 2) and filament orientation with respect to the load that modulates the pushing force (*yellow arrows*, factor 3). *Grey dots*: actin monomers and subunits. *Grey gradient*: gradient of actin monomers (darker color for higher ($C$)). *Red dots*: transient actin attachment to the load through the NPFs—Arp2/3 complex—actin filament ternary complex (see also related Supplementary Figs 1 and 2). *Scale bars* in **a**, **d** are 15 μm. *Scale bars* in **b**, **e**, **f** are 10 μm

the growth rate of LMs and the thin actin tails, respectively. Based on these results and previous experimental and theoretical work[5, 22–24], we formulated a minimal mathematical equation

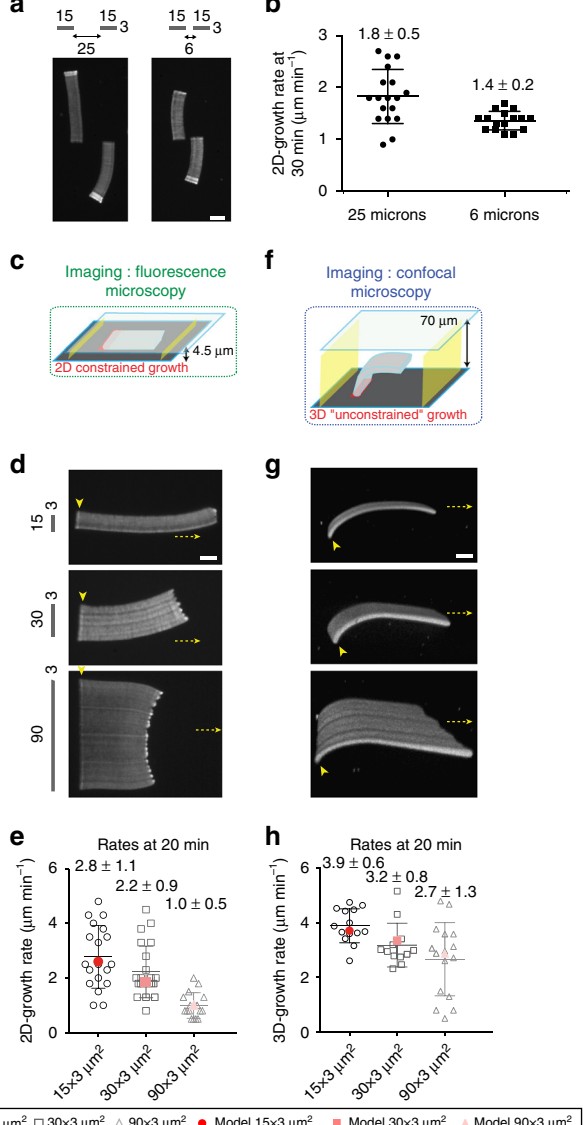

**Fig. 2** Actin assembly induces a local monomer depletion that controls the growth rate of branched actin networks. **a** LMs were constrained as in **c** and grown from $3 \times 15\,\mu m^2$ NPF-coated bars either spaced by 25 or 6 μm. **b** The growth rate was measured at 30 min. *Error bars* show mean s.d. for *n* LMs per condition, *n* = 18 from four experiments (25 μm), *n* = 16 from six experiments (6 μm). **c–f** Actin assembly was followed for LMs growing from NPF-coated nucleation bars of 3 μm width and increasing length, as mentioned. The growth was followed for 2D-like configuration in small volume with 4.5 μm space between glass and coverslip **c**, **d**, **e**, or for 3D-like configuration in a larger reconstituted volume with 70 μm space between glass and coverslip **f**, **g**, **h**. **e**, **h** 2D- and 3D-growth rates measured in each case (*open black symbols*). Mathematical modeling (*red symbols* in **e**, **h**) of the experimental growth rates measured in **d**, **g**. *Error bars* show mean s.d. for *n* LMs, *n* = 19 from four experiments (15 μm), *n* = 23 from seven experiments (30 μm), and *n* = 16 from four different experiments (90 μm) in **e** and *n* = 14 from six experiments (15 μm), *n* = 12 from six experiments (30 μm), and *n* = 16 from six experiments (90 μm) in **h**. **d**, **g** LM growth was reconstituted with the standard purified medium containing actin-profilin complex, Arp2/3 complex and capping protein, as described in the Methods section. All specified lengths are in μm (see also related Supplementary Figs 3, 4, 5 and 6). *Scale bars* in are 15 μm

(the actin network growth-rate model) that best described the growth rate of LMs (Fig. 1h). In this equation, the network growth rate is a function of: (i) the barbed end rate of assembly ($k_{on}$); (ii) the monomer half-size ($\delta$); (iii) the local monomer concentration at the nucleation site, which was calculated by solving a set of differential equations for monomer diffusion and assembly using an experimentally determined diffusion coefficient (Supplementary Fig. 3 and Supplementary Methods); and (iv) a geometry/mechanical factor that resists against network growth. This latter geometry/mechanical factor $\Phi$ integrates the impacts of (i) the angle of actin filaments impinging the nucleation site (geometry/architecture factor), and (ii) the transient tethering[22] of de novo nucleated filaments (mechanical factor) (Fig. 1h and Supplementary Methods). Mathematical estimates of network-growth rates using this equation showed unambiguously that the effect of local monomer depletion was negligible for thin tails formed on an NPF-spot (Fig. 1b) compared those formed on broad LMs (Fig. 1a). This allowed for a direct experimental measurement of the geometry/mechanical factor that resists against network growth: $\Phi = V_{\text{thin tail}} / V_{\text{free polymerization}} = 0.7$ (Supplementary Methods). Given our expectation that the network architectures and dynamics of actin tethering were the same in both thin tails and broad LMs, the value of $\Phi$ was kept constant. Thus, according to the actin network growth-rate model, the significant slowdown of the growth rate measured for broad LMs compared with the thin actin tails was due to either a local decrease in monomer concentration because of monomer consumption at the site of active assembly, or due to mechanical friction in the actin network, or due to both.

**Monomer depletion occurs by assembly at the nucleation site.** To distinguish between the effects of actin monomer depletion or mechanical friction on network growth rates, we compared the growth rates of two similar but physically independent networks when they grew distant (25 μm apart) or proximal (6 μm apart) from each other (Fig. 2a and Supplementary Movie 3). We observed a significant drop in the growth rate of the two proximal networks compared with the two distal networks (Fig. 2b). One explanation for this drop is that the two distal networks (Fig. 2a, two bars separated by 25 μm) use monomers from two separate areas around them, whereas the two proximal networks use monomers from areas that overlap (Fig. 2a, two bars separated by 6 μm) and therefore the overlapping areas lead to a higher depletion in the local monomer concentration at the site of active nucleation.

To further explore the relationship between the size of the nucleation area and the extent of local monomer depletion, we analyzed the growth rates of LM of different widths generated on bar-shaped patterns of increasing size (15, 30 and 90 μm) in "2D" (Fig. 2c, d and Supplementary Movie 4). The growth rate of the LMs decreased as the size of the nucleation area increased (Fig. 2d, e, *black symbols*). The processing of the data using the actin network growth-rate model (Fig. 2e, *red symbols*) and keeping the geometry/mechanical $\Phi$ factor constant for these LMs (Supplementary Methods) revealed that reduced growth rate was due to a greater depletion of monomers at the site of active nucleation of wider LMs.

To confirm the above relationship, we calculated the local concentration of actin monomers at the nucleation sites when the LMs were assembled in a 2D configuration. We solved equations for monomer diffusion and actin assembly based on the controlled parameters of our reconstituted systems (Supplementary Figs 3 and 4a, b, Supplementary Methods[33]). The solutions to the equations revealed that actin assembly at the nucleation site

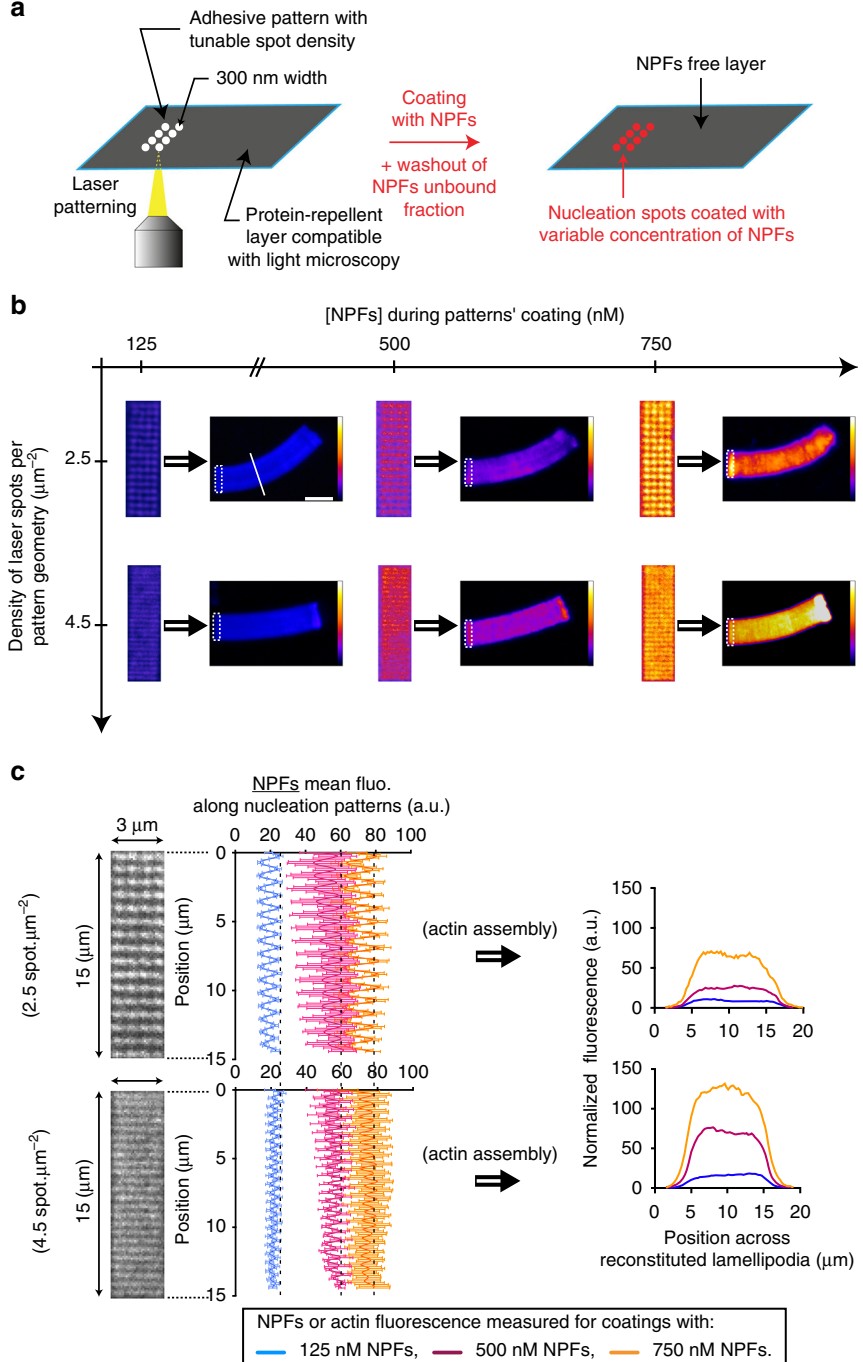

**Fig. 3** Laser-patterning method allows for the precise control of the density and the geometrical organization of actin filaments in growing actin network. **a** Adhesive spots of 300 nm in diameter were printed on silane-PEG-coated coverslip with 355 nm pulsed laser. Adhesive patterns consisted of arrays of spots of controllable density, and were specifically coated with a defined amount of NPFs. **b** Two independent parameters, spot density of the nucleation pattern and NPF concentration used for pattern coating, controlled actin filament density and geometrical organization within LMs. Dynamic LMs were reconstituted with the standard purified medium. For each image, *dotted rectangles* indicate the position of the nucleation pattern, and the LUT scale measures the actin-network fluorescence. **c** Quantitative analysis of NPF density coated on the nucleation patterns. For each coating condition, the NPF fluorescence on $3 \times 15\,\mu m^2$ patterns was measured and the mean fluorescence extracted. After actin assembly on patterns, actin fluorescence was measured across LMs (along the *white line* as indicated in panel **b**, *top left* image). *Error bars* represent standard deviation (see also related Supplementary Fig. 7). *Scale bar* is 15 μm

led to a strong depletion of actin monomers, not only in the local vicinity of the leading edge of LMs, but in the significant volume surrounding it (Supplementary Fig. 4a, b). Moreover, we found that this depletion effect depended on the size of the nucleation site (Supplementary Fig. 5a, b), and on the distance between nucleation sites (Supplementary Fig. 6). After 20 min of actin assembly, monomer concentration drastically dropped to 32, 22, 12% of the initial concentration for the 15, 30 and 90 μm patterns, respectively, in a 5 μm-wide border around the nucleation site (Supplementary Fig. 5a, b). By processing these data using the actin network growth-rate model and using the parameters described in Fig. 1, the relationship between local monomer

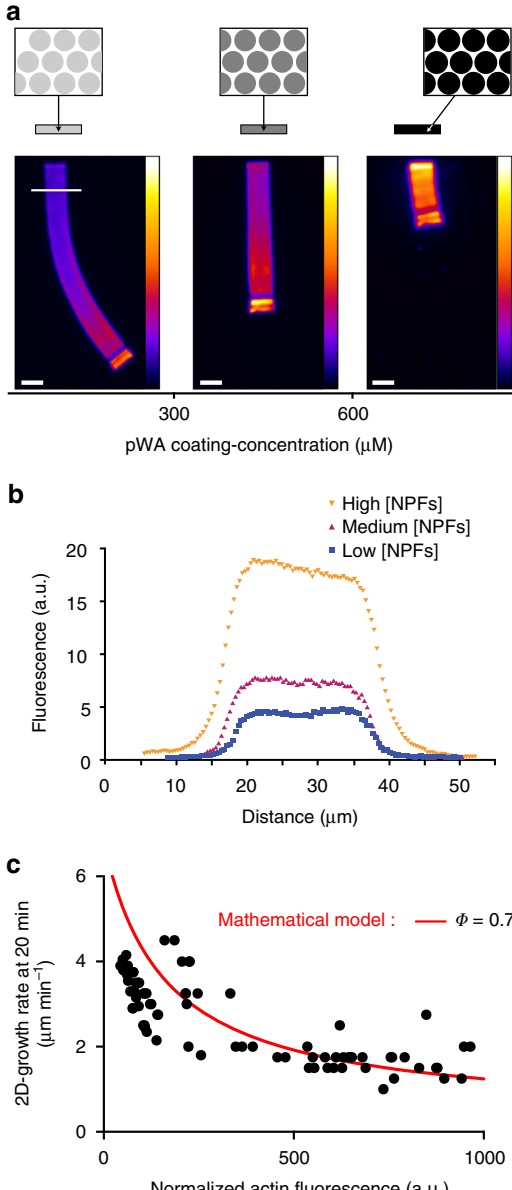

**Fig. 4** Actin filament density controls the growth rate of branched actin network. **a** Images of typical LMs of variable filament density reconstituted on bar-shaped patterns of constant geometry and variable NPF concentration. The images taken after 78 min of actin assembly reveal that denser networks grew slower than the sparser ones. **b** The network density was measured across the LMs (along the *white line* as indicated in **a**, *left image*). **c** Network growth rates at 20 min were represented as a function of the network fluorescence. Simulated growth rates at 20 min (*red solid line*) were calculated as a function of computed filament densities and according to the measured parameter $\Phi = 0.7$ (Fig. 1; Supplementary Methods). *Scale bars* are 15 µm (see also related Supplementary Fig. 8)

depletion and nucleation area were quantitatively and accurately simulated (Fig. 2e, compare experimental data (*black symbols*) with simulated values (*red symbols*)). The observed depletion of actin could not have been the result of global actin depletion. Indeed, given the steady-state cumulative length of LMs, our assays contained a total amount of actin monomers approximately 13 orders of magnitude larger than the number of actin subunits assembled in the F-actin networks. Similarly, quantitative estimates show that the consumption of Arp2/3 complex or capping protein by the networks were not significant enough to deplete the local concentrations of these molecules (Supplementary Methods). Thus, our results demonstrated that the sustained assembly at the nucleation site established diffusive gradients that led to local monomer depletion.

To further validate the monomer-depletion hypothesis, we extended our model to consider diffusive monomer gradients in a 3D configuration (Supplementary Fig. 4c, d). The model predicts that the monomer flow towards the nucleation site should be higher in 3D than in 2D (12-fold in the case of Fig. 2f compared with Fig. 2c, Supplementary Methods). In agreement with the hypothesis, the local monomer depletion was less prominent around the nucleation site in the 3D configuration (Supplementary Fig. 4c, d and Supplementary Fig. 5c, d) than in the 2D configuration (Supplementary Fig. 4a, b and Supplementary Fig. 5a, b) and the observed LM growth rates for the patterns were significantly higher in the 3D configuration than in the 2D configuration (Supplementary Fig. 5e, and compare *black open symbols* Fig. 2e, h). Note that the relationship between local

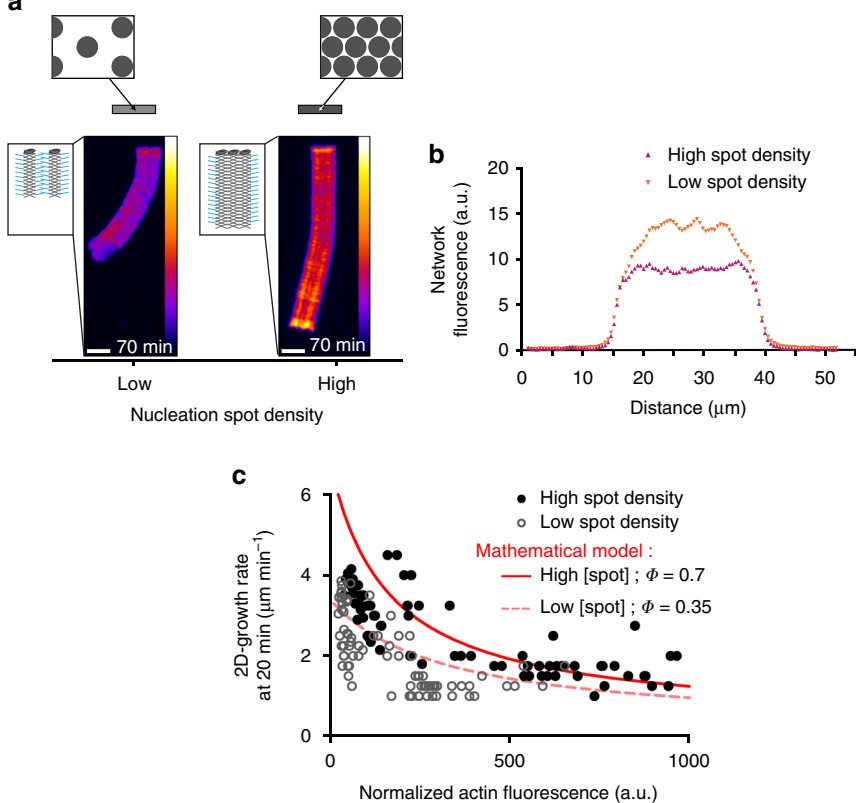

**Fig. 5** Actin filament organization modulates the growth of the network. **a** Images of typical LMs of variable NPF-spot density on bar-shaped pattern. The images taken after 70 min of actin assembly reveal that in contrast to Fig. 4, sparser networks grew slower than the denser ones. **b** The network density was measured across the LMs (along the *white line* as indicated in Fig. 4a, *left* image). **c** Network growth rates at 20 mn were represented as a function of the network fluorescence. Experimental data are represented by the symbols (high spots density, *closed circles*; low spots density, *opened circles*). Simulated growth rates at 20 min (*red and dashed solid lines*) were calculated as a function of computed filament densities according to a 2-fold degrease in $\Phi$ between the high- and low-spot density patterns. *Scale bars* are 15 μm (see also related Supplementary Fig. 8)

monomer depletion and nucleation area held in the 3D configuration (Fig. 2h). Importantly, the results obtained from the comparison between the 2D and 3D configurations argue against a strong effect during LM growth of the friction of the filaments against the wall of the experimental chamber. On the contrary, the difference in growth rates between 2D and 3D can be fully accounted for by the difference in the local monomer-depletion effect, without changing the geometry/mechanical factor in the equation for the growth rate. Hence, we reasoned that increasing viscosity by adding methylcellulose would reduce monomer diffusion but should have a minimal effect on the mechanical friction (Supplementary Methods). In this regime and as expected, LM growth rate was slightly lower with higher viscosity (Supplementary Fig. 5f).

**A new experimental setup to control growing actin network**. To uncouple the contribution on actin growth of the orientation of filaments within the network from the positions at which they are tethered within the nucleation site, we developed a novel and versatile experimental method that allows the precise control of the spacing between the nucleation spots (i.e. spot densities) in the nucleation area (Fig. 3). To this end, we used a pulsed UV laser to print nucleation patterns that consist of arrays of nucleation spots of a predefined density (Fig. 3a and Supplementary Movie 5). We only used spot densities that led to the reconstitution of continuous LMs on the patterns. As the branching reaction is confined to the surface of the nucleation spot (300 nm in diameter) and the actin filaments extend outside

the spot (reflecting the continuous aspect of the LMs), we hypothesized that the distance between spots controls both the orientation of actin filament within the network and the density of filaments tethering in the nucleation area. Hence, by varying the density of the nucleation spots and/or the amount of NPFs grafted to these spots, we were able to modulate the geometrical organization and density of an actin network (Fig. 3b). For every spot density evaluated, the amount of NPFs grafted per nucleation spot remained constant for each given NPF concentration (Fig. 3c and Supplementary Fig. 7). Accordingly, the density of the spots correlated well with the concentration of NPFs (Supplementary Fig. 7). Moreover, the density of spots correlated well with the fluorescence intensity of the LMs (Fig. 3c). Therefore, this method appeared suitable to fine-tune the overall filament organization and density of the LMs.

**Actin filament density modulates LM growth rate**. Using the method described above, we explored how actin-filament density controlled the growth rate of the branched networks (Fig. 4a and Supplementary Movie 6). Using patterns with high-spot densities and various concentrations of NPFs, LMs were generated with different filament densities, which were quantified by assessing their fluorescence intensity as function of the NPF concentration (Fig. 4b). We analyzed the LM growth rate as a function of LM fluorescence, which in turn is dependent on filament density (Fig. 4c). Our results showed that LMs generated with low-NPF concentrations contained lower densities of filaments and had higher growth rates than the LMs generated by higher NPF

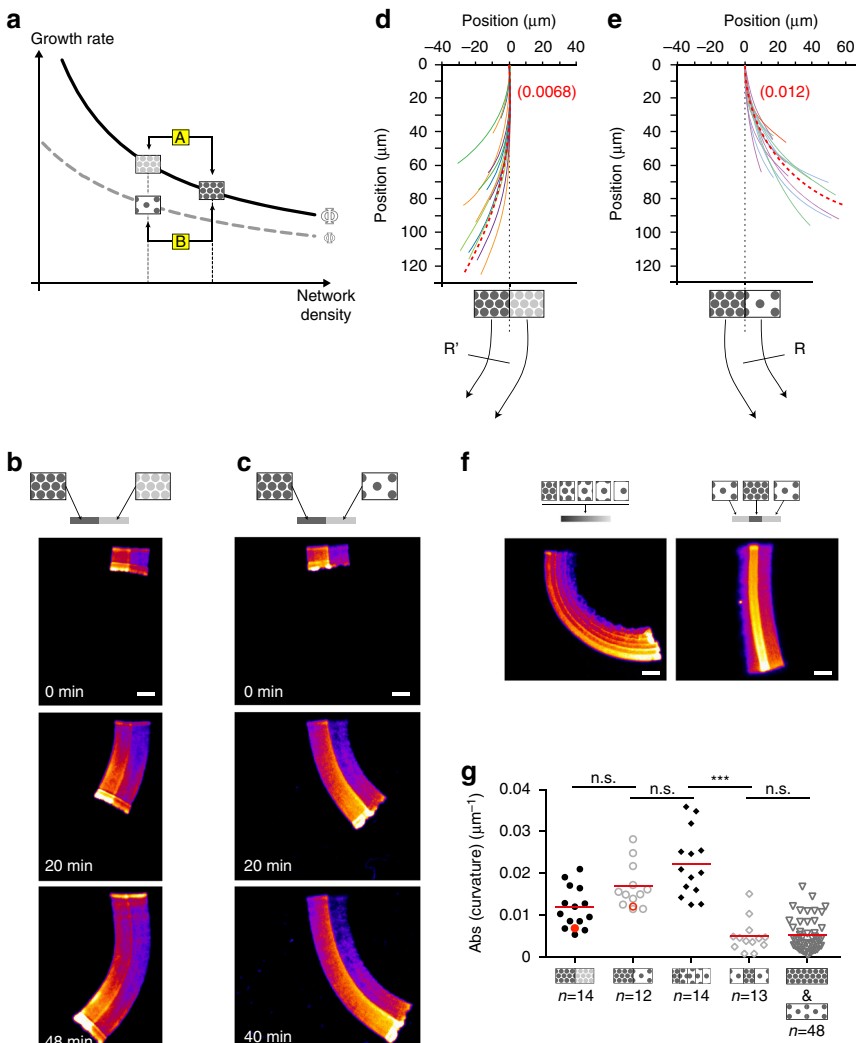

**Fig. 6** Density and geometrical organization interplay to control the steering of heterogeneous networks. **a** Cartoon predicting heterogeneous actin-network-growth behavior. If the two densities are generated by the same pattern family (i.e. two positions along the same curve; Case A), the steering is predicted to be always towards the denser network; if the two densities are generated by two different pattern families (two positions from two different curves Case B), the steering can be towards the sparser network. *Solid curve* refers to networks generated by *high*-spot-density patterns; *dotted curve* refers to networks generated by *low*-spot-density patterns. Double-density branched actin networks were polymerized on heterogeneous patterns. Patterns consisted either in an array of spots of the same density, with one half coated with high and the other half with low amounts of NPFs per spot **b**, or in an array of spots of two distinct densities both coated with the same amount of NPFs per spot **c**. Network curvatures, represented by the traces of the boundary between high- and low-density sides, were reported at 40 min of assembly in **d**, **e**, referring to **b**, **c** respectively. The *red dashed line* in **d**, **e** represents the predicted curvature according to the model. **f** Complex heterogeneous patterns were generated. A gradient of high- to low-spot density (*left panel*) or a symmetric and alternate low-high-low spot density (*right panel*). **g** LMs were polymerized on nucleation zones as indicated, and their curvature measured. High and low densities were 6.6 and 2.5 spots $\mu m^{-2}$, respectively. Gradually sparse patterns were a series of $3 \times 5\ \mu m^2$ *rectangles* of decreasing spot density (from 8.3 to 2 spots $\mu m^{-2}$). The "low-high-low" patterns were a series of $3 \times 10$, $3 \times 6$, $3 \times 10\ \mu m^2$ *rectangles* of 2.5, 6.6, 2.5 spots $\mu m^{-2}$, respectively. ***$p < 0.001$, multiple comparison Šídák method. *Scale bars* are 15 $\mu$m (see also Supplementary Fig. 9)

concentrations (Fig. 4c, *black dots*). This result was consistent with the local monomer depletion hypothesis, in that the LMs with higher filament densities will consume more actin monomers than those with lower filament densities. Processing the data through the actin network growth-rate model using the parameters derived from the preceding experiments, the decrease in the LM growth rate was satisfactorily simulated as a function of the increased density of the network (Fig. 4c, *red line*). We therefore concluded that for nucleation areas, which have the same spot densities, the growth of higher filament density networks leads to higher local monomer depletion, which in turn slows down the network-growth rate.

**Actin network architecture dependence of LM growth rate**. We then hypothesized that actin filament arrays tethered by nucleation spots were more effective at developing pushing forces from elongation than non-tethered actin arrays (see cartoons with schematic network growth in Fig. 5a). To address this hypothesis, we compared the growth rate of LMs generated by two different (low and high) densities of spots in the same-sized nucleation area and with the same NPF concentration per spot (Fig. 5a and Supplementary Movie 7). We confirmed that the distance between the nucleation spots controlled the density of LMs (quantified by LM fluorescence; Fig. 5b). Unexpectedly and seemingly in contradiction with the above results (Fig. 4), LMs

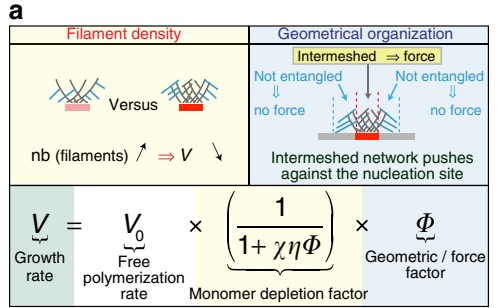

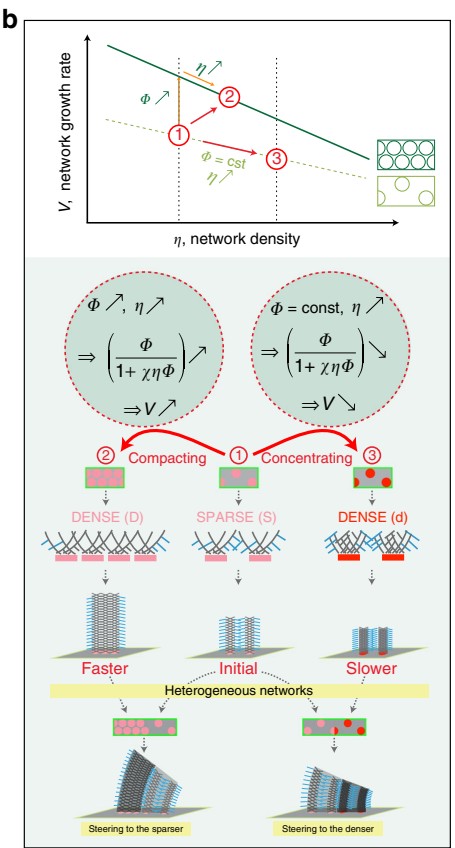

**Fig. 7** Mechanochemical model for growing branched actin network. **a** Network growth rate (*green box*) can be summarized by three terms; (i, *white box*) $V_0$, free polymerization rate of barbed ends; (ii, *yellow boxes*) the monomer depletion factor, which is inversely proportional to ($\chi \times \eta \times \Phi$), where $\chi$ represents monomer consumption due to polymerization corrected by diffusion, $\eta$ is the network density, and $\Phi$ is the network geometry factor of 0.7 calculated in Fig. 1; and, (iii, *blue boxes*). As depicted in *yellow boxes*, because soluble proteins were maintained constant throughout the experiments, higher nucleation density (*pink* to *red*), translates into higher filament density and yields higher local monomer depletion, thus lower growth rate. *Blue boxes* summarize the influence of the geometry factor according to which filaments push more efficiently against the load when they are in contact with the NPF area. Accordingly, actin filaments not in contact with the nucleation area do not produce as much force and are more prone to bending and/or capping. **b** Rationale for the control of speed and steering during actin network growth according to two scenarios based on actin filaments density and network architecture. The first is, from (1) to (2), where actin networks evolved by a geometrical reorganization of the network at a constant network density, or from (1) to (3) where actin networks evolved by maintaining the geometry of the network constant but increasing its density. As depicted in cartoons, for the compacting networks—the (1) to (2) case—as the network changes from low (1) to high (2) organization, the density is constant but the efficiency of the network pushing against the load is higher. Therefore, the actin network in (2) is growing faster than the network in (1). For concentrating networks—the (1) to (3) case—individual networks generated by nucleation surface unit (*pink* to *red dots*) become denser, without changing their organization. Therefore, the actin network in (3) is growing slower than the network in (1). According to these simple rules, heterogeneous networks will steer (or turn) towards the less growth-efficient actin network overall

generated by nucleation areas with a high spot density had an overall growth rate 1.3-fold greater (statistically significant) than LMs generated by nucleation areas with a low spot density. Interestingly, when these growth rates were plotted as a function of LM filament density (i.e. actin fluorescence intensity; Fig. 5c), the growth rates for LMs with identical filament densities were greater for LMs generated by nucleation areas with a high-spot

density than for those generated by nucleation areas with a low-spot density (Fig. 5c, *black dots* above *open dots*). We attributed the lower growth rate with nucleation areas of low spot density (Fig. 5c) to the contribution of the geometrical organization (architecture) of actin filaments (see cartoons in Fig. 5a). We hypothesized that LMs comprised two sub-populations of actin filaments: a population that is more effective for protrusion (i.e.

developing pushing forces) because the filaments are tethered at NPFs spots; and a population that is less effective for protrusion because the filaments are situated between NPFs spots. Based on this hypothesis, we assumed a 2-fold decrease in the geometry/mechanical factor $\Phi$ between LMs generated by low vs. high spot density nucleation areas. Using this assumption in the actin network growth-rate model, the LM growth rates were satisfactorily simulated (Fig. 5c, *red dotted and solid lines*; Supplementary Fig. 8). Accordingly, we concluded that the LM-growth rate is dependent on filament density—via the extent of local monomer depletion—and on filament orientation and tethering that controls the efficiency by which filaments develop pushing forces and hence protrusions.

**Density and architecture control steering of actin network**. To investigate how the local regulation of LM growth rates could impact the steering of protrusions, we generated heterogeneous LMs made of different nucleation-spot densities. We tested two different conditions to manipulate the heterogeneity of the growing actin network. First, the two halves of the nucleation area differed in the NPF concentrations per nucleation spot but had the same nucleation-spot density (Fig. 6a, Case A and Supplementary Movie 8 *left*); and second, the two halves of the nucleation area differed in nucleation-spot densities but had the same NPF concentration per spot (Fig. 6a, Case B and Supplementary Movie 8 *right*). In Case A (Fig. 6a), we predicted that when the nucleation architecture is constant ($\Phi$ is constant), the filament density within the network controls the growth rate as a function of the local monomer concentration. Therefore, the denser side of the network depletes more monomers than the sparser side (Supplementary Fig. 9a–d). In accordance with the prediction, the growth rate at the side of the nucleation area with the denser network of actin filaments was lower than at the side with a sparser network of actin filaments and, the overall direction of network growth was deflected (i.e. steered) towards the denser side (Fig. 6b, d).

In Case B and according to our prediction, the geometry/mechanical factor $\Phi$ would control the growth rate as a consequence of the different nucleation spot densities (Fig. 6a, c). Indeed, the network growth rate at the side with the lower spot density was lower than that at the side with the higher spot density, and hence the growth of the heterogeneous network steered towards the side with a lower spot density (Fig. 6c, e and Supplementary Fig. 9e–h). Moreover, the side with the lower spot density also generated a filament density that was lower than at the side with the higher spot density. Therefore, these results show that the direction of network growth can be modulated by the architecture of the branched network, in addition to and concomitant with the density of the actin filaments. To further examine this general rule about steering control during LM growth, we evaluated more complex patterns of nucleation areas consisting of a graded density of nucleation spots (Fig. 6f, *left panel* and Supplementary Movie 9) or of a central area of high-spot density surrounded by two areas of low-spot density (Fig. 6f, *right panel*). Interestingly, the actin growth-rate model could account quantitatively for the steering of this complex actin network (Fig. 6g, *red symbols* and Supplementary Fig. 9d, h). Therefore, we conclude that the density of the nucleation spots and the resulting architecture of the branched actin network determine the growing properties of LMs and emerge as critical factors in controlling the steering of LM growth.

## Discussion

This study has established how the heterogeneity in a branched actin network can control its growth rate and the orientation of this growth (i.e. steering). Specifically, by combining experimental observation and theoretical modeling, we have demonstrated that actin-monomer depletion and the architecture of the actin filaments at the site of assembly are critical to this control during LM growth. Therefore, we propose that the fine-tuning of these two parameters within the cell enable a diversity of branched actin network growth behaviors that are fundamental to controlling cell motility and its steering.

The most dramatic effect on the rate of growth of the experimental actin networks was obtained when the size and/or the NPF density of the nucleation area were increased (Figs 1a, b, 2d, 4a, 5a). How can these two related variables affect the rate of growth? At the point of contact with the patterned NPFs, actin-filament nucleation and elongation consume rapidly the available local pool of actin monomers. This generates a local depletion of available monomers slowing down filament elongation and therefore the growth rate of the LM. A variation in the density of filaments in contact with the nucleation area will have therefore a direct effect on the LM growth rate via monomer depletion (Fig. 7a, "Filament density"). Indeed, a local increase of NPF concentration tends to generate a greater local depletion of monomers and thus to locally slow down filament elongation forcing the direction of network growth overall to turn towards such regions where the filament density is high (Fig. 7b, "concentrating" scenario). According to this description of actin-based motility, the dynamic localization of actin monomers will provide a potential spatiotemporal mechanism to regulate the protrusion efficiency during cell locomotion. This view is supported by early theoretical work[33] and a recent in vivo study on neuronal motility [34]. In this latter study, the modulation of the expression of thymosin β4, a monomer sequestering protein, regulates the local pool of actin monomers at the leading edge of the cell and the underlying LM protrusion and growth cone motility[34]. To maximize the protrusion, cells may locally adopt a denser and more homogeneous distribution of the nucleation promoting complexes (Fig. 7, solid curve in the plot of $V$ vs. $\eta$), ensuring thus an optimal filament density, leading to optimal network stiffness in order to resist the membrane tension and induce protrusion, but with a limited effect on the local monomer depletion.

Our results demonstrate that the NPF distribution at the site of nucleation directly impacts LM growth rate (Fig. 5). We propose that two populations of actin filaments are present in contact with the site of nucleation (Fig. 7a "Geometrical organization"). One population that is effective at force production during LM growth because it contains actin filaments transiently tethered with NPF spots[22]; and a second population that is not effective at force production because it contains actin filaments present between NPFs spots and not directly tethered to them[22]. Indeed, a local increase of NPF packing will generate a denser network with a higher pushing efficiency, forcing the network to turn away from the region where the filaments are tethered and potentially at high density (Fig. 7b, "compacting" scenario). The dependence of the rate of growth with the architecture of actin branched network is consistent with the relationship between LM architecture and protrusion behavior[3, 35]. However, in the cellular context the contribution of actin filaments within the LM generated by additional factors including formins or ENA/VASP, introduces another level of complexity in the regulation of protrusion speed and force generation[36, 37].

Chemotaxis and haptotaxis cues as well as signaling feedback loops are known to either promote or silence Arp2/3 complex-mediated branching during the steering of cell motility[3, 6, 7]. In the case of haptotaxis, cells can sense differences in extracellular matrix (ECM) composition and modulate their Arp2/3 complex-dependent nucleation to adapt and migrate up the ECM gradient[7, 38]. An explanation on how these signals may act on the

LM organization to control steering comes from the fact that these inputs can modulate the amount of NPFs as well as their distribution along the membrane, leading to networks that are more or less efficient at protruding. Accordingly, the heterogeneity of the actin network can control steering during cell motility depending on the filament densities within the network and the degree of membrane tethering, and is sufficiently responsive to enable the cell to adjust its motility in a changing environment. Therefore, our actin growth-rate model provides a general framework to describe how the steering is controlled during cell locomotion and how this is an emergent property of the heterogeneity of actin networks in the LM.

## Methods

**Protein production and labeling**. Actin was purified from rabbit skeletal-muscle acetone powder[39]. Monomeric Ca-ATP-actin was purified by gel-filtration chromatography on Sephacryl S-300[40] at 4 °C in Buffer G (5 mM Tris-HCl [pH 8.0], 0.2 mM ATP, 0.1 mM CaCl₂ and 0.5 mM dithiothreitol (DTT)). Two grams of muscle acetone powder were suspended in 40 ml of buffer G and extracted with stirring at 4 °C for 30 min, then centrifuged 30 min at 30,000×g at 4 °C. The supernatant with actin monomers was filtered through glass wool and we measured the volume. The pellets were suspended in the original volume of Buffer G and we repeated the centrifugation and filtration steps. While stirring the combined supernatants in a beaker add KCl to a final concentration of 50 mM and then 2 mM MgCl₂ to a final concentration of 2 mM. This step will polymerize the actin monomers. After 1 h, add KCl to a final concentration of 0.8 M while stirring in cold room. This dissociates any contaminating tropomyosin from the actin filaments. After 30 min, centrifuge 2 h at 140,000×g to pellet the actin filaments. Discard supernatant and gently wash off the surface of the pellets with buffer G. Gently suspend the pellets in about 3 ml of buffer G per original gram of acetone powder using a Dounce homogenizer and dialyze for 2 days vs. three changes of buffer G to depolymerize the actin filaments. To speed up depolymerization, you can sonicate the suspended actin filaments gently. Clarify the depolymerized actin solution by centrifugation in Ti45 rotor at 140,000×g for 2 h to remove aggregates. The top 2/3 of the ultracentrifuge tube contains "conventional" actin. Gel filter on Spectral S-300 in buffer G to separate actin oligomers.

Actin was labeled on lysines with Alexa-568[41]. Labeling was done on lysines by incubating actin filaments with Alexa568 succimidyl ester (Molecular Probes). All experiments were carried out with 5% labeled actin.

The Arp2/3 complex was purified from bovine thymus[42]. Take a calf thymus from −80 °C and put it in a water bath at room temperature. Meanwhile, add protease inhibitors to 200 ml of Arp2/3 complex extraction buffer (20 mM Tris pH 7.5, 25 mM KCl, 1 mM MgCl₂, 5% glycerol). In the cold room, cut the thymus in ~1 cm pieces. Blend it in 100 ml extraction buffer for 1–2 min. Pour the extract into a beaker and stir it for 30 min. Spin the extract in a tabletop centrifuge at 1700×g for 5 min and then spin the clarified supernatant at 39,000×g for 25 min at 4 °C. Filter the supernatant through glass wool. Carefully set pH to 7.5 with KOH (try not to overshoot). Spin for 1 h at 140,000×g at 4 °C. Take the middle aqueous phase and transfer it to a chilled glass beaker. Precipitate the extract with 50 % ammonium sulfate. Spin at 39,000×g at 4 °C for 30 min. Suspend the pellet in 10 ml extraction buffer with 0.2 mM ATP, 1 mM DTT and protease inhibitor. Dialyze overnight against Arp2/3 dialysis buffer (20 mM Tris pH 7.5, 25 mM KCl, 1 mM MgCl₂, 5 % glycerol, 1 mM DTT and 0.2 mM ATP). Make a GST-WA glutathione sepharose column and wash it with the extraction buffer with 0.2 mM ATP, 1 mM DTT and protease inhibitor. Run the dialyzed extract over the GST-WA. Wash the column with 20 ml extraction buffer with 0.2 mM ATP, 1 mM DTT. Wash the column with 20 ml extraction buffer with 0.2 mM ATP, 1 mM DTT and 100 mM KCl. Elute the Arp2/3 complex with 20 ml extraction buffer with 0.2 mM ATP, 1 mM DTT and 200 mM MgCl₂. Dialyze the Arp2/3 complex in source A buffer (piperazine-N,N′-bis(2-ethanesulfonic acid) (PIPES) pH 6.8, 25 mM KCl, 0.2 mM ethylene glycol-bis(β-aminoethyl ether)-N,N,N′,N′-tetraacetic acid (EGTA), 0.2 mM MgCl₂ and 1 mM DTT) overnight. Spin the protein at 1700×g for 5 min. Add KCl to a final concentration of 975 mM to make 500 ml of source B buffer. Load the Arp2/3 complex on MonoS column and elute with source B buffer. Dialyze the Arp2/3 complex into storage buffer (10 mM Imidazole pH 7.0, KCl 50 mM, MgCl₂ 1 mM, ATP 0.2 mM, DTT 1 mM and glycerol 5%), flash frozen in liquid nitrogen and stored at −80 °C.

GST-WA, GST-pWA[43] are expressed in Rosettas 2 (DE3) pLysS. Fusion protein was purified by glutathione–Sepharose affinity chromatography (Amersham) and stored in Buffer PWA (20 mM Tris pH 8, 150 mM NaCl, 1 mM DTT, 0.5 mM ethylenediaminetetraacetic acid (EDTA)). Human profilin[44] is expressed in BL21 DE3 pLys S *Escherichia coli* cells. Culture is grown in LB medium + 100 µg ml⁻¹ carbenicilin to OD of 0.6 at 600 nm, then 0.5 mM isopropyl β-D-1-thiogalactopyranoside (IPTG) is added and cultures are grown for four more hours at 37 °C. Pelleted cells are resuspended in Buffer P (20 mM Tris pH 8.0, 150 mM KCl, 0,2 mM DTT, 1 mM EDTA) + 2 M Urea. Following sonication and centrifugation the clarified extract is loaded on a polyproline sepharose column equilibrated in buffer 1 + 2 M Urea. Resin is washed with four volumes of buffer P

+ 3 M Urea. Profilin is eluted with Buffer P + 8 M Urea. Pooled fractions are dialyzed extensively to remove urea in storage buffer (20 mM Tris pH 8.0, 1 mM EDTA, 1 mM DTT). Protein is centrifuged at 150,000×g for 30 min to remove precipitate. Protein aliquots are stored at 4 °C for 6 weeks, or flash frozen in liquid nitrogen and stored at −80 °C, and mouse CP (α/β)[45] is cloned in a pRFSDuet-1 plasmid (Novagen) containing two cloning sites. The full length CP is a 6× His tagged at the N-terminus of the α subunit. CP is expressed in Rosetta2 DE3 pLys S in LB carbenicillin (100 µg ml⁻¹). Culture is grown until OD is 0.6 at 600 nm. Induction is achieved by addition of 0.5 mM IPTG at 26 °C overnight. Cells are pelleted and suspended in Buffer CP (20 mM Tris pH 8.0, 250 mM NaCl, 10 mM Imidazole, 5% Glycerol, 1 mM DTT, 1 mM EDTA) + protease inhibitors cocktail tablet. Cells are then sonicated and centrifuged at 39,000×g. Supernatant is applied to 1 ml of Ni sepharose fast flow resin (GE Healthcare). After 1 h at 4 °C under gentle rotation, resin is washed with 20 volumes of Buffer CP containing 20 mM Imidazole. Elution is performed with buffer CP + 300 mM Imidazole. Purified protein is dialyzed overnight against a storage buffer (20 mM Tris pH 8.0, 1 mM DTT, 1 mM EDTA, 0.2 mM CaCl₂), flash frozen in liquid nitrogen and stored at −80 °C. GST-pWA constructs attached to glutathione beads were labeled by incubating 1 ml of a 50% resin suspension overnight at 4 °C with 7 excess molar ratio of Alexa-488 (Molecular Probes) in TBSE (10 mM Tris-HCl pH 8.0, 100 mM NaCl and 1 mM EDTA)[46].

**Nanoablation station**. Inverted microscope (TE2000-E, Nikon) equipped with a CFI S-Fluor oil objective (×100, NA 1.3, Nikon), a perfect focus system (Nikon), motorized stage (Marzhauser), and a dual-axis galvanometer that focalizes the laser beam on the sample on the field of the camera, including a telescope that adjust the laser focalization with the image focalization, and polarizer to control the laser power (iLasPulse device, Roper Scientific). The microscope uses a pulsed laser passively Q-switched laser (STV-E, TeamPhotonics) that delivers 300 ps pulses at 355 nm (energy per pulse 1.2 µJ, peak power 4 kW, variable repetition rate 0.01–2 kHz, average power ≈ 100 mW). The laser was scanned throughout the region of interest, ROI, with a power set to 300 nJ. The ROIs (or patterns) used in this study were rectangles of usually 3 µm width and 15 µm long or as indicated. The laser displacement that defines the laser spot density, the distance between the patterns, and the number of repetition of patterned rectangles, as well as the laser exposure time were controlled using Metamorph software (Universal Imaging Corporation).

The microscope is moreover equipped with a fluorescence illumination system X-Cite 120PC Q (Lumen Dynamics) and QuantEM:512SC camera (Photometrics) to monitor the laser printing procedure.

**Functionalization of laser-patterned surfaces**. 20 × 20 mm² coverslips and cover glasses (Agar Scientific) were extensively cleaned, oxidized with oxygen plasma (3 mn, 30 W, Harrick Plasma, Ithaca, NY, USA) and incubated with 1 mg ml⁻¹ of Silane-PEG overnight. Patterns of the desired area were printed on Silane-PEG-coated surfaces using the nanoablation station.

For patterns homogeneously coated with the same concentration of NPFs, immediately after laser-patterning patterned coverslips were coated with a solution of the nucleation promoting factor GST-pWA at the appropriate concentration (typically between 100 and 1000 nM) for 15 min[47]. When needed, the fluorescence density of the NPFs density was quantified before the assembly of actin on patterns.

For patterns of the same spot density but with two concentrations of NPFs, half of the patterns were printed with a 6.6 spot⁻¹µm⁻², coated with 300 nM GST-pWA, the excess of GST-pWA was wash out, and the surface was dried. The same procedure was then repeated to print the second pattern halves on the coverslips with pre-coated halves, the second round of coating was performed with 300 nM GST-pWA, the excess of GST-pWA was washed out and the surface was dried, ready to assess actin assembly[47].

**Bead coating**. Carboxylate polystyrene microspheres (2 µm diameter, 2.6% solids-latex suspension, Polysciences, Inc) were mixed with 2 µM GST-pWA in X buffer (10 mM 4-(2-hydroxyethyl)-1-piperazineethanesulfonic acid (HEPES) [pH 7.5], 0.1 M KCl, 1 mM MgCl₂, 1 mM ATP, and 0.1 mM CaCl₂) for 15 min at 20 °C on thermoshaker. The beads coated with GST-pWA were then washed in X buffer solution containing 1% bovine serum albumin (BSA) and stored on ice for 48 h in X buffer-0.1% BSA. GST-pWA surface density on the beads was quantified on SDS-PAGE gel: 2.4 × 104 pWA µm⁻².

In order to control the reconstitution chamber height, we used BSA-coated 4.5 µm carboxylate polystyrene microspheres (4.5 µm diameter, 2.6% solids-latex suspension, Polysciences, Inc) as pillars. Briefly, beads were incubated for 15 min at 20 °C on thermoshaker in X buffer solution containing 1% BSA, then pelleted and stored in ice for 48 h in X buffer-0.1% BSA.

**Reconstituted LMs assembly and bead motility assay**. Assembly of reconstituted LMs was either performed in small or large volume of the polymerization medium in polymerization chambers of 20 × 20 mm² × 4.5 or 70 µm height, respectively (Figs 1 and 2). The actin polymerization mix containing 6 µM actin monomers (5% Alexa568 labeled), 18 µM profilin, 120 nM Arp2/3, 25 nM CP, in X buffer (10 mM HEPES [pH 7], 0.1 M KCl, 1 mM MgCl₂, 1 mM ATP, and 0.1 mM

CaCl$_2$) supplemented with 1% BSA, 0.2% methylcellulose, 3 mM DTT, 0.13 mM 1,4-diazabicyclo[2.2.2]octane (DABCO), 1.8 mM ATP, 0.02‰ red fluorescent beads (0.2 μm, 2% solids suspension, 580/605, Molecular Probes), 0.008% BSA-coated 4.5 μm beads in the case of LMs reconstitution in a small polymerization volume, and 0.008% pWA-coated 4.5 μm beads in the case of comparison between LMs assembly and actin-based bead motility.

To normalize actin network fluorescence between assays we used in the polymerization medium 0.2 μm fluorescent beads (Molecular Probes), at a dilution allowing for the presence of around 10 tiny beads per observation field. The network fluorescence at a given time of assembly was the average fluorescence measured in a $5 \times 5 \, \mu m^2$ ROI in the LMs at 10 μm from the nucleation pattern edge. For each polymerization assay, the maximum fluorescence of beads was then taken as a reference to normalize network fluorescence.

Growth rate were calculated using ImageJ software. The 2D-growth rate at a given time was calculated according to the network elongation during the last 4 min. When the LMs were elongated in a large reconstitution volume and grew in the Z-direction, we used the Simple Neurite Tracer plugins of ImageJ that allows for the visualization of the image stack through the XZ, ZY and XY planes. Points taken along the LM trace in the Z-stack at the proximal and the distal LM edges permit the calculation of the LM length. Thus, the 3D-growth rate at a given time $t$ was then calculated according the elongation of LMs (total length $t$ minus total length at $t-2$ min) during the last 2 min. We use the view through the XY plane to calculate network fluorescence as described above.

**Image acquisition.** For the 2D growth of reconstituted LMs, image acquisition was performed using an upright Axioimager M2 Zeiss microscope equipped with an EC Plan—Neofluar dry objective (×20, NA 0.75), a computer controlled fluorescence microscope light source X-Cite 120PC Q (Lumen Dynamics), a motorized XY stage (Marzhauser) and an ORCA-ER camera (Hamamatsu). For the 3D growth of reconstituted LMs, image acquisition was performed using an Eclipse TI-E Nikon inverted microscope equipped with a CSUX1-A1 Yokogawa confocal head, an Evolve EMCCD camera (Roper Scientific), a CFI Plan APO VC oil objective (×60/NA 1.4; Nikon), a CFI Plan Fluor oil objective (×40/NA 1.3 and ×100/1.45; Nikon), and a motorized stage MS 2000 (ASI imaging). Both stations were driven by MetaMorph software (Universal Imaging Corporation). The use of the motorized stage allowed acquiring actin dynamics of several networks assembled either on beads or on micropatterns under exactly the same biochemical conditions.

**Mathematical model.** The modeling is based on numerical solutions of diffusion–reaction partial differential equations for G-actin distribution and of algebraic equations for balancing fluxes. The details are in the Supplementary Methods.

**Code availability.** Numerical codes used to solve the reaction–diffusion equations describing actin monomer distributions can be downloaded from: http://cims.nyu.edu/~mogilner/codes.html

**Data availability.** The data that support the findings of this study are available from the corresponding authors on reasonable request.

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

## Acknowledgements

This work was supported by grants from Human Frontier Science Program (RGP0004/2011 awarded to L.B.), Agence Nationale de la Recherche (MaxForce, ANR-14-CE11-0003-01 awarded to L.B.), National Institute of Health Grant (GM068952 awarded to A.M.) and ERC starter Grant (310472) to M.T. R.B.-P was supported by the Institut Universitaire de France.

## Author contributions

R.B.-P., A.M., M.T., L.B., C.S. designed the research; R.B.-P., C.S., T.K., C.G. performed the research; R.B.-P., C.S., T.K., J.Z., C.G., A.M., M.T., L.B. analyzed the data; J.Z., A.M. wrote the mathematical model, R.B.-P., A.M., M.T., L.B. wrote the paper.

## Additional information

**Competing interests:** The authors declare no competing financial interests.

