## [Peer Review File · Nature Communications]

Reviewers' comments:

Reviewer #1 (Remarks to the Author):

This is a very interesting paper that analyses effects on growth velocity of branched actin networks nucleated from custom-engineered areas of substratum-attached activators of Arp2/3 complex (e.g. bars of pWA) in vitro. Interestingly, such pWA-coated bars when incubated with actin, profilin, Arp2/3 complex and capping protein generate Arp2/3 complex-dependent actin networks that grow from the substratum-associated pWA-bar and push the developing actin network rearwards, in analogy to Arp2/3-dependent lamellipodia that continuously grow from the tip of the lamellipodium and protrude by pushing of the membrane forward solely counteracted by concomitant rearward translocation of the actin network, which is several fold larger in extent than protrusion in most cell types. Considering all these analogies, the authors have decided to term the in vitro-generated structures "lamellipodium-like structures" (LMs), which seems well justified, at least concerning these dynamic considerations. Using this system and playing with observations on network growth velocity influenced by size and distances between individual, independently generated LMs, the authors make two major observations. First, growth velocity of these networks is influenced by amount of actin monomer capable of diffusing to sites of growth. In other words, increased densities of these networks can reduce network velocities simply through monomer depletion. However, an additional parameter in their networks can also influence growth velocity, which is the structural/geometrical organization of the network. Using elegant variations of pWA concentrations versus pattern geometries (i.e. density of pWA-coated spots), the authors convincingly show that networks of increased densities can also grow more efficiently if the areas of F-actin regions (assumed not to be entangled in between pWA-coated regions) are decreased. All these observations are backed up by mathematical modelling. All this makes this story quite complete, and I feel that it should essentially be published as is, although I would recommend modest tuning down of the wording concerning the relevance for lamellipodial actin network turning in vivo. More specifically, although the authors clearly show how monomer depletion in their system and geometrical considerations of their networks can influence local growth velocities, and thus network turning, it does remain unclear, of course, what precisely is going on in cells during specific turning events, which is much more difficult to determine. It should also remain clear in the text that the two described parameters recognized to influence network growth velocities are just two out of many potential parameters that could influence turning in vivo, and it is unclear, in fact, how relevant the parameters affecting growth here will be relevant in vivo. As an example, to what extent are cells capable of modulating the homogeneity of Arp2/3 complex activation along the leading edge? What would be the precise in vivo correlate of regions of high versus low Arp2/3 activity within a given, protruding lamellipodium? I believe that it will probably exist but the reader should at least be notified that answers to such questions are not yet fully established in vivo.

Related to this topic, there is actually one clear error in the discussion at the end of the top paragraph on page 12: The last sentence of the chapter reads: "...the dependence of the rate of actin-based movement with the density of actin branched network is consistent with faster cell motility for denser actin filament networks within lamellipodia (ref 35, which is Bear et al., 2002). Actually, the opposite was described and observed in reference 35, as this work deals with effects of Ena/VASP proteins on branched actin filament networks, and the argument was back then that enhanced Ena/VASP activities elongate filaments leading to lower density of branched actin networks and thus FASTER protrusion. This is also summarized in ref. 3 (Krause and Gautreau, 2014). In Figure 5 of this review by Krause and Gautreau, at the right-hand side, it is summarized how decreased (not increased!) density of the Arp2/3-dependent network, as effected by the activity of Ena/VASP family proteins for instance, causes increased protrusion and thus network velocity, at least in the authors' view. So this argument has to be corrected.

Aside from this necessary correction, and without particularly asking for additional experiments that I find obligatory for publication, I would like to ask a few (partly experimental) questions that could help readers judge more easily the relevance of this work for Arp2/3-dependent actin networks in vivo.

Specific comments:

1.) I feel that it will be relevant to discuss the dimensions of lamellipodia versus LMs studied in vitro here. In particular, the observed monomer depletion effect should be dependent, at least to certain extent, on relative sizes of actin networks versus availability of assembly-competent actin monomer. More specifically, the actin networks generated are pretty large as compared to lamellipodia, and actin monomer concentrations perhaps comparably low as compared to what's available inside cells (although precise numbers in vivo are admittedly controversial). However, most striking for me was the thickness of generated actin networks, which seems to correlate with the thickness of the bars used for substrate coating (3 μ m in many cases). It seems that this causes actin networks of thickness with 4 μ m or more (as seen e.g. in Figure 2), which is about 30x thicker than average lamellipodia, which are roughly 100-150nm thick. So my question is: what is the relevance of thickness for the monomer depletion effect? Can this be extracted from mathematical modelling and/or can the authors generate much thinner bars and show how this affects monomer depletion? In other words, it would be great if the authors could confirm that monomer depletion might also be relevant for smaller, and in particular much thinner, actin networks such as those found in lamellipodia.

2.) All experiments were performed in the absence of actin disassembly factors such as ADF/cofilin family members, the presence of which might eliminate or at least modify the monomer depletion effect, as expected from in vitro work published previously (e.g. Loisel et al., 1999 and many studies thereafter). Can the authors show a simple experiment addressing how ADF/cofilin activity might modify this parameter and/or at least discuss how they feel actin filament disassembly within the network might affect the relevance of the monomer depletion parameter?

3.) This concerns the Φ (Phi)-parameter (geometric / force factor) that is tuned by the density of pWA-coated spots. The authors argue that in regions in between entangled filament regions, pushing is less efficient because filaments are more prone to capping and/or bending (see for instance legend to Fig. 8A), but would it be possible to show (perhaps using just a thin line of coated spots) that regions in between entangled, pWA-coated filament networks do indeed contain less Arp2/3 and more capping protein? I am also not so sure whether in these regions filaments will really bend more, perhaps they will just slide on the surface and bend less because of being incapable of pushing efficiently? So the wording could be amended in this context. Whatever the case, it might be helpful to back up some of these assumptions with some more experimental data, such as specific labelling of the mentioned network components, perhaps using antibodies or fluorescently-labelled, purified components if possible.

Minor comments:

4.) I am not so sure how relevant Figure 3 is to the average reader, so this could perhaps go to the Supplement.

5.) It would be worth proof-reading the text for correct grammar. I am not listing every error, but in many instances, for example, the authors use plural form where it should be singular, such as "two-colors experiments" (page 3), which should correctly read two-color experiments, "...adequate surfaces nucleation patterns..." (page 4), which should read adequate surface nucleation patterns, "...NPFs concentrations..." (page 8), where it should read NPF concentrations, "...had the same nucleation spots density..." (page 9), which should read nucleation spot density, "...during LMs growth..." (page 10), which should correctly read LM growth. In the legend to Figure 1, it should read "...same as..." instead of "...same than..."; in the legend to Figure 8, the "0" in V_0 should be subscript and the text should read "...pushing against the load..." instead of "...again the load..." (bottom of page 18).

6.) In addition, in the text and Figure legend, the authors use the greek letter "Eta" (η), but in the Figures (e.g. Figure 8), the symbol has been converted to a simple n, which makes things difficult

to follow; this should be corrected where appropriate.

7.) In Figure 4C, it seems a bit odd that actin intensities are expressed as "actin tail fluorescence" in labeling of the ordinate, and the measured distance through the structure is labelled as "reconstituted lamellipodia". The authors should use consistent nomenclature for their structures in order to avoid confusion of readers.

8.) In the legend to Figure 7A, the authors are describing curve cases and I assume mean to say curve cases "A" and "B" (see Figure) instead of cases "B" and "C". This should also be corrected.

9.) Figure S1A shows what happens concerning actin assembly on these bar structures in the absence of capping protein with two colors of actin added consecutively, which is very nice, but the authors should consider stressing/emphasizing in the text that in both elongation (at the periphery of the structure) and in the nucleation+elongation zone in the center of the structure, the authors are looking at barbed end assembly, because readers not entirely familiar with these systems might assume pointed elongation at the periphery perhaps, which is of course not possible in these conditions.

Reviewer #2 (Remarks to the Author):

This manuscript by Boujemaa-Paterski and colleagues uses an elegant in vitro system to unravel fascinating new properties of how to control actin-based motility. In this work, they can control the density and spatial distribution of nucleating proteins on the surface of a glass coverslip and then add the requisite purified proteins to control the assembly of a reconstituted lamellipodia – the actin-based active gel that drives protrusion of the leading edge of motile cells. This is extraordinary and beautiful data – the experimental results are quite clear and definitive. and appropriate for publication in Nature Communications. In addition, the authors have collaborated with Alex Mogilner who has developed a theory to understand these findings, namely to describe how local monomer depletion can result of a lower protrusion rate. Less clearly described (but clear in the data) is why the spatial distribution of NPF can also control growth rate by a "compaction" method. My comments below are suggestions to the authors on how they might improve the clarity of the manuscript. However, I think the quality of this work is quite high and certainly deserves to be published in nature communications.

1) The choice of the word "constrained networks" is a bit odd to me. A word that more directly reflects "limited" or "restricted" filament length might be preferably. I immediately think of mechanical constraints.

2) I find the description of the surprising results in Figure 6 confusing. The experimental results are quite clear, although I guess it would be nice if the authors had a few more perturbations to test their idea. Perhaps it would help if there was a schematic of what the authors are thinking.

3) In a related point, I found the box plot in Fig. 6C not so useful – 6D was very clear and sufficient

4) Related to point 2, the concepts of 'tethered', 'geometrical organization', 'compacting', 'pushing efficiency' and 'efficient force production' are not made clear

5) (In Discussion) What are the mechanisms of haptotaxis and chemotaxis regulating NPF?

6) Can the authors give an estimate of the differences in the forces generated under these conditions to give an idea of how this mechanism could be used for turning? Are the scales used here relevant to migratory cells?

7) In motile cells, protrusive fronts are typically seen as traveling wave and are transient. Except in keratocytes, it is unclear how important the mechanism described here is important for cell turning. Can the authors comment on this?

8) What are the ranges of monomer concentration/NPF density that the authors think are

relevant- are there estimates of whether these are those that occur in motile cells?

Reviewer #3 (Remarks to the Author):

This paper presents measurements of actin network growth on small beads and bars that are micropatterned with pWA, an actin filament nucleation promoting factor (NPF). It is found that monomer depletion has strong effects, including the following: i) networks on beads grow faster than on bars, ii) networks on small bars grow faster than on large bars, iii) networks growing from different sources are slowed by each other's monomer consumption, and iv) reducing the pWA surface concentration, without changing the spatial pattern, increases the growth rate. However, when the global pWA concentration is instead reduced by lowering the density of nucleation spots (keeping a fixed NPF density in each one), the growth rate is reduced. The observations of monomer depletion are supported by calculations of diffusion profiles.

The importance of monomer depletion for the system studied is an important result for the field. It is convincingly argued by the combination of experiments and diffusion calculations, both of which use well-established and reliable methods. Depletion effects have been predicted by several previous theoretical papers and invoked to explain different types of measurements, but this is the most solid demonstration and complete exploration to date.

However, the relevance to biological cells is not sufficiently clear:

a) How large are the depletion effects expected to be in cells, on the basis of existing estimates of quantities such as filament density? In cells, the free-actin concentration is much higher and most of the actin is sequestered - how does this affect the extent of monomer depletion?

b) It was found that increasing the NPF concentration reduces the network growth speed. In cells, the concentration of active NPFs can be increased by local chemical signals or localized photoactivation of Rho GTPases such as Rac. Are the authors suggesting that such localized activation would reduce the speed of polymerization in cells? This would seem to contradict the finding that localized Rac activation causes protrusion (Wang X., He L., Wu Y.I., Hahn K.M., Montell D.J. *Nat Cell Biol.* 2010;12:591-597.)

Minor issues:

a) The discussion of the unexpected effect of increasing the spot spacing is focused on force generation, but the total force on the actin network resulting from viscous drag should be very small. So it is not clear that the relative force-generating capacities of tethered and untethered filaments are the most important effect. An alternative hypothesis is that filaments "reaching out" from the NPF spots become tethered to areas between the spots, without growing. The tethered filaments would hold the growing filaments back. I would suggest that the authors explore a broader range of hypotheses in discussing this effect.

b) In the regime studied by the authors, the growth rate increases with dropping NPF concentration, provided that the distribution is not changed. But this must stop at some point, since at zero NPF concentration there is no network growth. Is there a crossover point where the plot of growth rate vs NPF concentration changes sign? If so, what would determine this crossover point?

c) One would assume that the barbed ends of the filaments are oriented toward the NPFs. Have the authors verified this, or do they have strong arguments for this assumption?

Response to reviewers:

We would like to thank all three reviewers for their positive opinions on our manuscript and for their excellent comments. We have now revised our manuscript according to their comments. We feel that the modifications asked by the reviewers strengthen our conclusions and improve the manuscript.

Reviewer #1,

This is a very interesting paper that analyses effects on growth velocity of branched actin networks nucleated from custom-engineered areas of substratum-attached activators of Arp2/3 complex (e.g. bars of pWA) in vitro. Interestingly, such pWA-coated bars when incubated with actin, profilin, Arp2/3 complex and capping protein generate Arp2/3 complex-dependent actin networks that grow from the substratum-associated pWA-bar and push the developing actin network rearwards, in analogy to Arp2/3-dependent lamellipodia that continuously grow from the tip of the lamellipodium and protrude by pushing of the membrane forward solely counteracted by concomitant rearward translocation of the actin network, which is several fold larger in extent than protrusion in most cell types. Considering all these analogies, the authors have decided to term the in vitro-generated structures “lamellipodium-like structures” (LMs), which seems well justified, at least concerning these dynamic considerations. Using this system and playing with observations on network growth velocity influenced by size and distances between individual, independently generated LMs, the authors make two major observations. First, growth velocity of these networks is influenced by amount of actin monomer capable of diffusing to sites of growth. In other words, increased densities of these networks can reduce network velocities simply through monomer depletion. However, an additional parameter in their networks can also influence growth velocity, which is the structural/geometrical organization of the network. Using elegant variations of pWA concentrations versus pattern geometries (i.e. density of pWA-coated spots), the authors convincingly show that networks of increased densities can also grow more efficiently if the areas of F-actin regions (assumed not to be entangled in between pWA-coated regions) are decreased. All these observations are backed up by mathematical modelling. All this makes this story quite complete, and I feel that it should essentially be published as is.

We would like to thank this reviewer for the very positive feedback on our manuscript.

Although I would recommend modest tuning down of the wording concerning the relevance for lamellipodial actin network turning in vivo. More specifically, although the authors clearly show how monomer depletion in their system and geometrical considerations of their networks can influence local growth velocities, and thus network turning, it does remain unclear, of course, what precisely is going on in cells during specific turning events, which is much more difficult to determine. It should also remain clear in the text that the two described parameters recognized to influence network growth velocities are just two out of many potential parameters that could influence turning in vivo, and it is unclear, in fact, how relevant the parameters affecting growth here will be relevant in vivo. As an example, to what extent are cells capable of modulating the homogeneity of Arp2/3 complex activation along the leading edge? What would be the precise in vivo correlate of regions of high versus low Arp2/3 activity within a given, protruding lamellipodium? I believe

that it will probably exist but the reader should at least be notified that answers to such questions are not yet fully established in vivo.

We have modified the manuscript accordingly to specify that we have determined two key parameters in controlling network growth and how these parameters can modulate steering of heterogeneous networks during motility but other parameters can also play a role.

Related to this topic, there is actually one clear error in the discussion at the end of the top paragraph on page 12: The last sentence of the chapter reads: "...the dependence of the rate of actin-based movement with the density of actin branched network is consistent with faster cell motility for denser actin filament networks within lamellipodia (ref 35, which is Bear et al., 2002). Actually, the opposite was described and observed in reference 35, as this work deals with effects of Ena/VASP proteins on branched actin filament networks, and the argument was back then that enhanced Ena/VASP activities elongate filaments leading to lower density of branched actin networks and thus FASTER protrusion. This is also summarized in ref. 3 (Krause and Gautreau, 2014). In Figure 5 of this review by Krause and Gautreau, at the right-hand side, it is summarized how decreased (not increased!) density of the Arp2/3-dependent network, as effected by the activity of Ena/VASP family proteins for instance, causes increased protrusion and thus network velocity, at least in the authors' view. So this argument has to be corrected.

We would like to thank the reviewer for this comment and have modified the text accordingly.

Aside from this necessary correction, and without particularly asking for additional experiments that I find obligatory for publication, I would like to ask a few (partly experimental) questions that could help readers judge more easily the relevance of this work for Arp2/3-dependent actin networks in vivo.

Specific comments:

1.) I feel that it will be relevant to discuss the dimensions of lamellipodia versus LMs studied in vitro here. In particular, the observed monomer depletion effect should be dependent, at least to certain extent, on relative sizes of actin networks versus availability of assembly-competent actin monomer. More specifically, the actin networks generated are pretty large as compared to lamellipodia, and actin monomer concentrations perhaps comparably low as compared to what's available inside cells (although precise numbers in vivo are admittedly controversial). However, most striking for me was the thickness of generated actin networks, which seems to correlate with the thickness of the bars used for substrate coating (3 μ m in many cases). It seems that this causes actin networks of thickness with 4 μ m or more (as seen e.g. in Figure 2), which is about 30x thicker than average lamellipodia, which are roughly 100-150nm thick. So my question is: what is the relevance of thickness for the monomer depletion effect? Can this be extracted from mathematical modelling and/or can the authors generate much thinner bars and show how this affects monomer depletion? In other words, it would be great if the authors could confirm that monomer depletion might also be relevant for smaller, and in particular much thinner, actin networks such as those found in lamellipodia.

We added section 'Relevance of the results to motile cells' in the Supplementary Information. We use modeling there to show that the actin monomer depletion effect is relevant for the leading edge lamellipodia. In short, the reason for the depletion effect to be in place for much thinner lamellipodial actin networks is: the rate of 'consumption' at the leading edge is proportional to the number of growing barbed ends, and thus, when the mesh size of the network is constant, to the thickness of the network. However, the diffusive flux bringing the monomers to the leading edge, is also proportional to the thickness of the experimental chamber or lamellipodium. Thus, the balance of the flux and consumption is independent of the thickness: in vitro, the network is ~ 10-20 times thicker, but the diffusive flux is also an order of magnitude greater than in thin lamellipodium. We provide mathematical details in the Supplementary Information; similarly, greater overall G-actin concentrations in motile cells are still accompanied by the depletion effect. Mathematics to back up this statement is more involved and is in the Supplementary Information.

2.) All experiments were performed in the absence of actin disassembly factors such as ADF/cofilin family members, the presence of which might eliminate or at least modify the monomer depletion effect, as expected from in vitro work published previously (e.g. Loisel et al., 1999 and many studies thereafter). Can the authors show a simple experiment addressing how ADF/cofilin activity might modify this parameter and/or at least discuss how they feel actin filament disassembly within the network might affect the relevance of the monomer depletion parameter?

We have an ongoing project that is addressing this question, but the role of ADF/cofilin on reconstituted lamellipodium is not simple and depends on the density of the actin network. This said, our preliminary data shows that the heterogeneous networks behave similarly in presence or absence of ADF/cofilin suggesting that even in presence of the disassembly machinery the local monomer depletion effect persists.

3.) This concerns the Φ (Phi)-parameter (geometric / force factor) that is tuned by the density of pWA-coated spots. The authors argue that in regions in between entangled filament regions, pushing is less efficient because filaments are more prone to capping and/or bending (see for instance legend to Fig. 8A), but would it be possible to show (perhaps using just a thin line of coated spots) that regions in between entangled, pWA-coated filament networks do indeed contain less Arp2/3 and more capping protein? I am also not so sure whether in these regions filaments will really bend more, perhaps they will just slide on the surface and bend less because of being incapable of pushing efficiently? So the wording could be amended in this context. Whatever the case, it might be helpful to back up some of these assumptions with some more experimental data, such as specific labelling of the mentioned network components, perhaps using antibodies or fluorescently-labelled, purified components if possible.

We tried the suggested experiments but the distance between spots and the resolution of the signal did not allow us to address this point. We have amended the text (see supplemental data) to address this comment.

Minor comments:

4.) I am not so sure how relevant Figure 3 is to the average reader, so this could perhaps go to the Supplement.

We have moved as suggested by the reviewer this figure to supplemental.

5.) It would be worth proof-reading the text for correct grammar. I am not listing every error, but in many instances, for example, the authors use plural form where it should be singular, such as “two-colors experiments” (page 3), which should correctly read two-color experiments, “...adequate surfaces nucleation patterns...” (page 4), which should read adequate surface nucleation patterns, “...NPFs concentrations...” (page 8), where it should read NPF concentrations, “...had the same nucleation spots density...” (page 9), which should read nucleation spot density, “...during LMs growth...” (page 10), which should correctly read LM growth. In the legend to Figure 1, it should read “...same as...” instead of “...same than...”; in the legend to Figure 8, the “0” in V_0 should be subscript and the text should read “...pushing against the load...” instead of “...again the load...” (bottom of page 18).

We thank this reviewer and have carefully proofread the manuscript.

6.) In addition, in the text and Figure legend, the authors use the greek letter “Eta” (η), but in the Figures (e.g. Figure 8), the symbol has been converted to a simple n, which makes things difficult to follow; this should be corrected where appropriate.

We have corrected the symbol in the figures.

7.) In Figure 4C, it seems a bit odd that actin intensities are expressed as “actin tail fluorescence” in labeling of the ordinate, and the measured distance through the structure is labelled as “reconstituted lamellipodia”. The authors should use consistent nomenclature for their structures in order to avoid confusion of readers.

We have corrected the figure 4C legend.

8.) In the legend to Figure 7A, the authors are describing curve cases and I assume mean to say curve cases “A” and “B” (see Figure) instead of cases “B” and “C”. This should also be corrected.

We have corrected the figure 7A legend.

9.) Figure S1A shows what happens concerning actin assembly on these bar structures in the absence of capping protein with two colors of actin added consecutively, which is very nice, but the authors should consider stressing/emphasizing in the text that in both elongation (at the periphery of the structure) and in the nucleation+elongation zone in the center of the structure, the authors are looking at barbed end assembly, because readers not entirely familiar with these systems might assume pointed elongation at the periphery perhaps, which is of course not possible in these conditions.

We have now addressed this issue in the text.

Reviewer #2,

This manuscript by Boujemaa-Paterski and colleagues uses an elegant in vitro system to unravel fascinating new properties of how to control actin-based motility. In this work, they can control the density and spatial distribution of nucleating proteins on the surface of a glass coverslip and then add the requisite purified proteins to control the assembly of a reconstituted lamellipodia – the actin-based active gel that drives protrusion of the leading edge of motile cells. This is extraordinary and beautiful data – the experimental results are quite clear and definitive. and appropriate for publication in Nature Communications. In addition, the authors have collaborated with Alex Mogilner who has developed a theory to understand these findings, namely to describe how local monomer depletion can result of a lower protrusion rate. Less clearly described (but clear in the data) is why the spatial distribution of NPF can also control growth rate by a “compaction” method.

My comments below are suggestions to the authors on how they might improve the clarity of the manuscript. However, I think the quality of this work is quite high and certainly deserves to be published in nature communications.

We would like to thank this reviewer for the very positive feedback on our manuscript.

1) The choice of the word “constrained networks” is a bit odd to me. A word that more directly reflects “limited” or “restricted” filament length might be preferably. I immediately think of mechanical constraints.

We have changed the word “constrained networks” to “restricted networks”

2) I find the description of the surprising results in Figure 6 confusing. The experimental results are quite clear, although I guess it would be nice if the authors had a few more perturbations to test their idea. Perhaps it would help if there was a schematic of what the authors are thinking.

We have included a scheme in figure 5 (old figure 6). We propose that the network’s architecture depends on the degree of homogeneity of the NPF distribution. If the PWA spots are spaced more sparsely, the actin network becomes less efficient in growth and force generation. One mechanistic possibility is – filaments’ angle distribution changes, as shown in the figure above, and so the same rate of elongation translates into slower growth of the network’s leading edge. Another possibility is that the balance between pushing and tethered filaments shifts so that relatively more filaments resist protrusion, and the growth slows. This and other plausible mechanisms are discussed in the Supplementary information.

3) In a related point, I found the box plot in Fig. 6C not so useful – 6D was very clear and sufficient

We have can removed this figure as suggested by this reviewer.

4) Related to point 2, the concepts of 'tethered', 'geometrical organization', 'compacting', 'pushing efficiency' and 'efficient force production' are not made clear

We improved the text to clarify these terms

5) (In Discussion) What are the mechanisms of haptotaxis and chemotaxis regulating NPF?

The exact mechanisms are not fully understood. We have suggested in the discussion a possible explanation based on our observations.

6) Can the authors give an estimate of the differences in the forces generated under these conditions to give an idea of how this mechanism could be used for turning? Are the scales used here relevant to migratory cells?

We are not actually suggesting that the turning mechanism is necessarily due to the differences in forces (it could be that differences in angular distributions within the actin network constitute the mechanism), though this is a possibility. If it is indeed the differences in forces, then those are in the pN per filament range, based on what we know about the polymerization force-velocity relation. In the migratory cells, there are no available direct measurements, but based on the existent measurements of the forces generated by a micron-long leading edge (nN range) and of the barbed end density (hundreds per micron), the in vitro forces are relevant to the forces in the migrating cells.

7) In motile cells, protrusive fronts are typically seen as traveling wave and are transient. Except in keratocytes, it is unclear how important the mechanism described here is important for cell turning. Can the authors comment on this?

The mechanisms of cell turning are being actively investigated, and from what we see in the literature (in the Supplementary Information in new section 'Relevance of the results to motile cells' we review this literature), these are multiple mechanisms. The mechanism we are proposing is certainly relevant for broad and steady lamellipodia, like those of keratocytes, but also, likely of nerve growth cones. In addition, there are conditions when fibroblasts move in the 'keratocyte'-like mode, with broad and steady leading edge. Not to forget, famous video on the internet of a crawling neutrophil chasing a bacterium made in the 1950s by the late David Rogers at Vanderbilt University shows the motile cell making turns by pivoting broad and steady leading edge in response to chemotactic signals. This said, we are certainly not proposing that ours is the dominant turning mechanism. We agree with the reviewer that often cells turn by either wave-like mechanism, or by extinguishing one local protrusion and building another in a new location (importantly, the turning can be governed also from the rear of the cell). We hope the reviewer agrees that it is important to unravel all possible turning mechanisms because each of them is likely to work in some physiologically relevant situation. Further discussion is in the Supplementary Information.

8) What are the ranges of monomer concentration/NPF density that the authors think are relevant- are there estimates of whether these are those that occur in motile cells?

The monomer concentration we use in the in vitro experiments is $\sim 6 \mu\text{M}$, which is likely the same as the concentration of polymerizable monomers in cells, because this concentration produces the polymerization rate of the order of that observed in vivo. It is likely, that total concentration of monomers in cells is much higher, and the major fraction is sequestered and unavailable for polymerization. In the Supplementary information we demonstrate that all our qualitative conclusions in that case remain the same. We are unaware of quantitative measurements of NPF density in motile cells, but what matter is, in fact, the actin network density (or mesh size). To measure it quantitatively in vitro would require probably EM, which is beyond the scope of our study. However, it is very likely (see arguments in the Supplementary Information) that this density in in vitro experiments is the same order of magnitude as that in the cells.

Reviewer #3:

This paper presents measurements of actin network growth on small beads and bars that are micropatterned with pWA, an actin filament nucleation promoting factor (NPF). It is found that monomer depletion has strong effects, including the following: i) networks on beads grow faster than on bars, ii) networks on small bars grow faster than on large bars, iii) networks growing from different sources are slowed by each other's monomer consumption, and iv) reducing the pWA surface concentration, without changing the spatial pattern, increases the growth rate. However, when the global pWA concentration is instead reduced by lowering the density of nucleation spots (keeping a fixed NPF density in each one), the growth rate is reduced. The observations of monomer depletion are supported by calculations of diffusion profiles.

The importance of monomer depletion for the system studied is an important result for the field. It is convincingly argued by the combination of experiments and diffusion calculations, both of which use well-established and reliable methods. Depletion effects have been predicted by several previous theoretical papers and invoked to explain different types of measurements, but this is the most solid demonstration and complete exploration to date.

We would like to thank this reviewer for the very positive feedback on our manuscript.

However, the relevance to biological cells is not sufficiently clear:

a) How large are the depletion effects expected to be in cells, on the basis of existing estimates of quantities such as filament density? In cells, the free-actin concentration is much higher and most of the actin is sequestered - how does this affect the extent of monomer depletion?

In the Supplementary Information, we added section 'Relevance of the results to motile cells' where we show using modeling that the depletion effect in lamellipodia of motile cells (and actually in other motile appendages as well) are supposed to be significant, including the case when the free-actin concentration is much higher and most of the actin is sequestered.

b) It was found that increasing the NPF concentration reduces the network growth speed. In cells, the concentration of active NPFs can be increased by local chemical signals or localized photoactivation of Rho GTPases such as Rac. Are the authors suggesting that such localized activation would reduce the speed of polymerization in cells? This would seem to contradict the finding that localized Rac activation causes protrusion (Wang X., He L., Wu Y.I., Hahn K.M., Montell D.J. Nat Cell Biol. 2010;12:591–597.)

We suggest that increasing the NPF concentration will decrease or increase the speed depending of the organization of the NPF (Figures 6 and 8, and see the ends of the ‘Actin monomer depletion and filament density in the protrusive network’ and ‘Geometrical organization of the nucleation site controls filament organization and network growth rate’ sections in the Discussion.). Thus, the finding that localized Rac activation causes protrusion is not inconsistent with our observations and theory.

Minor issues:

a) The discussion of the unexpected effect of increasing the spot spacing is focused on force generation, but the total force on the actin network resulting from viscous drag should be very small. So it is not clear that the relative force-generating capacities of tethered and untethered filaments are the most important effect. An alternative hypothesis is that filaments "reaching out" from the NPF spots become tethered to areas between the spots, without growing. The tethered filaments would hold the growing filaments back. I would suggest that the authors explore a broader range of hypotheses in discussing this effect.

We were not entirely clear in describing possible mechanisms explaining why the network's growth rate depends on the degree of homogeneity of the NPF distribution. What we observe is that if the spots are spaced more sparsely, the actin network becomes slower. Largely speaking, two, not mutually exclusive, effects can explain this: 1) change in average filament orientation, 2) change of mechanical balance between pushing and tethered filaments. One mechanistic possibility is change in network's architecture – filaments generated at the NPF spots bend or turn to reach and fill the spaces between the spots, so the angle distribution changes, and so the same rate of elongation of individual filaments translates into slower growth of the network's leading edge. Another possibility is that, as the reviewer suggests, the balance between pushing and tethered filaments shifts so that for some reason relatively more filaments get tethered between the NPF spots, the mechanical resistance to protrusion increases, and the network's growth slows down. There are also other possibilities – for example, larger spaces between the NPF spots could lead to lesser filament entanglement, which makes the actin network more deformable so that it recoils under load and slows down the protrusion. Finally, it is possible that when the NPF spots are sparse, the network becomes a weaved mesh of narrow actin tails which buckle and meander, again slowing the protrusion down. We added relevant discussion to the Supplementary Information.

b) In the regime studied by the authors, the growth rate increases with dropping NPF concentration, provided that the distribution is not changed. But this must stop at some

point, since at zero NPF concentration there is no network growth. Is there a crossover point where the plot of growth rate vs NPF concentration changes sign? If so, what would determine this crossover point?

This is a very good point. Indeed, for a very low NPF concentration, we expect the growth rate to stop increasing. There are three relevant factors: First, at a slow actin density (approximately an order of magnitude lower than in the in vitro experiments), the monomer depletion effect becomes negligible, and the growth rate becomes independent from the NPF concentration. Second, at even lower F-actin density, the external mechanical load, which may not scale with the number of pushing filaments, could overwhelm the network growth mechanically. Third, we show in the paper that the actin growth rate decreases when inhomogeneity of the actin network increases. By the law of large numbers, spatial inhomogeneity is an increasing function of the network density, so at very low density of NPFs, the growth rate will go down. We added a detailed discussion to the Supplementary Information in section 'Relevance of the results to motile cells'

c) One would assume that the barbed ends of the filaments are oriented toward the NPFs. Have the authors verified this, or do they have strong arguments for this assumption?

We have addressed in part this question in the Figure 1 and by the two colors experiments in Figure Supplemental 1. We can show that new actin assembly occurs at the pattern (Figure Supp1) and that capping proteins constrain this growth figure 1. This question was also addressed in Reymann et al., Nat. Mat. 2010, Supp Fig2.

REVIEWERS' COMMENTS:

Reviewer #2 (Remarks to the Author):

The authors have addressed all my concerns. I fully support publication.

Reviewer #3 (Remarks to the Author):

The authors have made appropriate changes, and the manuscript is now suitable for publication.